# Intercropping of *Cyphomandra betacea* with Different Ploidies of *Solanum* Sect. *Solanum* (Solanaceae) Wild Vegetables Increase Their Selenium Uptakes

**DOI:** 10.3390/plants12040716

**Published:** 2023-02-06

**Authors:** Lijin Lin, Xiangting Xu, Jin Wang, Xun Wang, Xiulan Lv, Yi Tang, Honghong Deng, Dong Liang, Hui Xia

**Affiliations:** 1Institute of Pomology and Olericulture, Sichuan Agricultural University, Chengdu 611130, China; 2College of Horticulture, Sichuan Agricultural University, Chengdu 611130, China

**Keywords:** *Cyphomandra betacea*, intercropping, *Solanum* sect. *Solanum*, selenium, antioxidant enzyme activity

## Abstract

Selenium (Se) deficiency causes various diseases in humans. Se can be obtained from fruits and vegetables. In this study, the fruit tree *Cyphomandra betacea* was intercropped with three *Solanum* sect. *Solanum* (Solanaceae) wild vegetables [diploid (*S. photeinocarpum*), tetraploid (colchicine-induced *S. photeinocarpum*), and hexaploid (*S. nigrum*)], respectively, and Se uptakes of these plants were determined by a pot experiment. Intercropping decreased the biomass, photosynthetic pigment content, and superoxide dismutase activity of *C. betacea*, but increased the peroxidase (POD) activity, catalase (CAT) activity, and soluble protein content of *C. betacea.* These indicators’ values of sect. *Solanum* increased after intercropping. The contents of Se increased in *C. betacea* and sect. *Solanum* after intercropping. Intercropped with diploid, tetraploid, and hexaploid increased the shoot Se contents in *C. betacea* by 13.73%, 17.49%, and 26.50%, respectively, relative to that of *C. betacea* monoculture. Intercropped with *C. betacea* increased the shoot Se contents in diploid, tetraploid, and hexaploid by 35.22%, 68.86%, and 74.46%, respectively, compared with their respective monoculture. The biomass and Se content of intercropped sect. *Solanum* showed linear relationships with the biomass and Se content of their monocultures. The biomass and Se content of intercropped *C. betacea* also exhibited linear relationships with that of sect. *Solanum* monocultures. Correlation and grey relational analyses revealed that the CAT activity, POD activity, and soluble protein content were the top three indicators closely associated with the *C. betacea* shoot Se content. The POD activity, soluble protein content, and translocation factor were the top three indicators closely associated with sect. *Solanum* shoot Se content. Therefore, intercropping can promote the Se uptake in *C. betacea* and sect. *Solanum* wild vegetables.

## 1. Introduction

Selenium (Se) is an essential trace element in humans. Humans obtain less than 40 μg d^−1^ of Se from the diet, and Se poisoning occurs after intake of concentration above 400 μg d^−1^ [1,2]. Se deficiency is associated with various human diseases such as heart disease, cancer, and reproductive disorders. High intake of Se in humans leads to loss of hairs and nails, and damage to the nervous and digestive systems [2]. The human body cannot synthesize Se; so, it is obtained through diet. Plant Se is the primary source of Se in humans [3,4]. Application of Se fertilization is a very common method for improving plant Se contents, but this approach is expensive with high environmental pollution risk [5]. Therefore, exploring effective and eco-friendly methods to improve Se accumulation in crops is imperative.

Intercropping can improve the rate of utilization of environmental resources by crops to some extent. Moreover, intercropping also can modulate the soil environment conditions, ultimately improving crop yield and quality to some extent [4,6]. The intercropping of maize with soybean is a typical representative of rational utilization of various resources for the production of crops [7]. Intercropping maize with legumes improved the absorption of nutrients and increased the yield of the two crops [8]. The heavy metal hyperaccumulator *Thlaspi caerulescens* intercropped with the non-hyperaccumulator *Thlaspi arvense* under heavy metal contaminated condition promoted the growth of the two plants, increased zinc (Zn) accumulation in *T. caerulescens*, and decreased Zn uptake in *T. arvense* [9]. Intercropping cadmium (Cd)-hyperaccumulator *Solanum photeinocarpum* or its post-grafting generations with loquat seedlings promoted the Cd accumulation and growth of the two plants [10]. In addition, intercropping of the Cd-hyperaccumulator *Galinsoga parviflora* with the Cd-accumulator plants *Capsella bursa-pastoris*, *Cardamine hirsuta*, and *Galium aparine* exhibited different effects on the Cd accumulation and growth of these plants, including promotion, inhibition, or no effects [11]. Three varieties (red, green, and black) of eggplant seedlings intercropped under Se-rich soil conditions showed increased Se accumulation and growth rate. Moreover, the three varieties exhibited increased photosynthetic pigment contents, antioxidant enzyme activities, and soluble protein contents [12]. The intercropping of three radish genotypes had various effects on their growth, physiology, and Se accumulation. Some intercropping combinations of radish promoted growth and Se accumulation of radish and improved their resistances to Se stress, whereas other intercropping combinations showed the opposite effects [13]. The intercropping of three genotypes of cherry tomato had various effects on the yield and quality of fruits, but all combinations showed increased Se contents in various organs [14]. These findings indicate that different intercropping combinations of crops may produce varying effects on their growth and Se accumulation, implying that appropriate intercropping combinations can promote Se accumulation in crops. Studies should explore suitable intercropping combinations to improve Se accumulation in crops.

*Cyphomandra betacea* is a self-pollinated perennial evergreen fruit tree with high edible and ornamental values, and there are only local varieties in China [15,16]. The Se accumulation capacity of *C. betacea* is lower than that of other Se-rich vegetables and fruits [17]. *Solanum photeinocarpum* and *Solanum nigrum* are annual to perennial *Solanum* sect. *Solanum* wild vegetables with a high Se accumulation capacity [18,19,20]. *S. photeinocarpum* is a diploid plant, and *S. nigrum* is a hexaploid plant, whereas *S. nigrum* is evolved from *S. photeinocarpum* in nature condition [21]. Intercropping *C. betacea* with *S. photeinocarpum* and *S. nigrum* may improve the Se accumulation capacities of these plants. Studies have not explored the effects of intercropping these plants on the Se accumulation capacities. Therefore, in this study, *C. betacea* was intercropped with *S. photeinocarpum* (diploid), colchicine-induced *S. photeinocarpum* (tetraploid), and *S. nigrum* (hexaploid), and the Se uptakes in these plants were evaluated to determine the best combination of sect. *Solanum* and *C. betacea* for improving the Se accumulation in these plants.

## 2. Results

### 2.1. Biomass of Plants

The root and shoot biomasses of *C. betacea* were lower than that of diploid, tetraploid, and hexaploid. The biomass order in different ploidies was diploid < tetraploid < hexaploid (Figure 1A,B). Compared with *C. betacea* monoculture, intercropped with diploid, tetraploid, and hexaploid decreased the root biomass of *C. betacea* by 6.07%, 15.63%, and 19.23%, respectively, and decreased the shoot biomass of *C. betacea* by 9.30%, 20.15%, and 23.13%, respectively. The root and shoot biomasses of different ploidies in intercropping were higher than that of their respective monoculture. In addition, the root biomasses of intercropped diploid, tetraploid, and hexaploid were linearly positively correlated with the root biomasses of their respective monoculture (r = 0.704, *n* = 9, *p* = 0.034, Figure 2A). The root biomasses of intercropped *C. betacea* were negatively correlated with the root biomasses of monoculture diploid, tetraploid, and hexaploid (r = −0.896, *n* = 9, *p* = 0.001, Figure 2B). The shoot biomasses of intercropped diploid, tetraploid, and hexaploid were positively related to the shoot biomasses of their corresponding monoculture (r = 0.922, *n* = 9, *p* = 0.000, Figure 2C). The shoot biomasses of intercropped *C. betacea* were linearly negatively correlated with the shoot biomasses of monoculture diploid, tetraploid, and hexaploid (r = −0.796, *n* = 9, *p* = 0.010, Figure 2D).

### 2.2. Photosynthetic Pigment Contents in Plants

Intercropped with diploid, tetraploid, and hexaploid showed lower contents of chlorophyll *a*, chlorophyll *b*, and carotenoid in *C. betacea* compared with the levels in *C. betacea* monoculture (Table 1). Notably, the chlorophyll *a*/*b* of *C. betacea* in intercropping was not significantly different with the level in monoculture. Intercropping showed no significant differences in the chlorophyll *a* and carotenoid contents in diploid relative to that of the diploid monoculture. The content of chlorophyll *b* in diploid was higher in intercropping, and the chlorophyll *a*/*b* of diploid was lower, compared with that of diploid monoculture. Tetraploid in intercropping showed higher contents of chlorophyll *a*, chlorophyll *b*, and carotenoid relative to that of tetraploid monoculture, whereas there was no significant difference in chlorophyll *a*/*b* between tetraploid intercropping and tetraploid monoculture. Hexaploid in intercropping exhibited higher levels of chlorophyll *a* and carotenoid compared with that in hexaploid monoculture. On the contrary, chlorophyll *b* content and chlorophyll *a*/*b* of hexaploid were not significantly different between monoculture and intercropping.

### 2.3. Antioxidant Enzyme Activities and Soluble Protein Contents in Plants

Intercropped with diploid and tetraploid did not exhibit significant differences in SOD activity of *C. betacea* compared with *C. betacea* monoculture (Table 2). Intercropped with hexaploid showed lower SOD activity of *C. betacea* than that of *C. betacea* monoculture. Intercropped with diploid, tetraploid, and hexaploid showed higher POD activity, CAT activity, and soluble protein content of *C. betacea* compared with the *C. betacea* monoculture. Diploid intercropped with *C. betacea* showed higher SOD activity, POD activity, CAT activity, and soluble protein content of diploid compared with diploid monoculture. Tetraploid intercropped with *C. betacea* did not exhibit significant differences in SOD activity of tetraploid compared with tetraploid monoculture. However, tetraploid intercropped with *C. betacea* had higher POD activity, CAT activity, and soluble protein content of tetraploid relative to tetraploid monoculture. Hexaploid intercropped with *C. betacea* did not exhibit significant differences in SOD activity and CAT activity of hexaploid relative to hexaploid monoculture. Compared with the hexaploid monoculture, intercropped with *C. betacea* had higher POD activity and soluble protein content of hexaploid.

### 2.4. Se Content and Translocation Factor of Plants

*C. betacea* intercropped with diploid, tetraploid, and hexaploid exhibited higher root and shoot Se contents compared with that of their respective monoculture (Figure 3A,B). Intercropped with diploid, tetraploid, and hexaploid increased the Se content in shoots of *C. betacea* by 13.73%, 17.49%, and 26.50%, respectively, compared with *C. betacea* monoculture. Intercropped with *C. betacea* increased the shoot Se contents in diploid, tetraploid, and hexaploid by 35.22%, 68.86%, and 74.46%, respectively, compared with their respective monoculture. In addition, the root Se contents in diploid, tetraploid, and hexaploid in intercropping were positively correlated with their root Se contents in monoculture (r = 0.760, *n* = 9, *p* = 0.018, Figure 4A). The root Se contents in *C. betacea* in intercropping were positively correlated with the root Se contents in diploid, tetraploid, and hexaploid in monoculture (r = 0.739, *n* = 9, *p* = 0.023, Figure 4B). The shoot Se contents in diploid, tetraploid, and hexaploid in intercropping were linearly positively correlated with their shoot Se contents in monoculture (r = 0.927, *n* = 9, *p* = 0.000, Figure 4C). The shoot Se contents in *C. betacea* in intercropping were linearly positively correlated with the shoot Se contents in diploid, tetraploid, and hexaploid in monoculture (r = 0.958, *n* = 9, *p* = 0.000, Figure 4D).

The findings showed that intercropped with diploid had higher TF of *C. betacea* relative to *C. betacea* monoculture (Figure 3C). Intercropped with tetraploid showed no significant difference in TF of *C. betacea*, whereas intercropped with hexaploid had lower TF of *C. betacea* compared with *C. betacea* monoculture. Diploid, tetraploid, and hexaploid in intercropping all exhibited higher TFs compared with their TFs in monoculture.

### 2.5. Relationship of Different Parameters in Plants

The root biomass and shoot biomass of *C. betacea* were highly significantly (*p* < 0.01) positively correlated with the chlorophyll *a* content, chlorophyll *b* content, carotenoid content, and SOD activity (Table 3). The root biomass and shoot biomass were highly significantly (*p* < 0.01) negatively correlated with the POD activity, CAT activity, and soluble protein content. The root Se content and shoot Se content were highly significantly (*p* < 0.01) positively correlated with the POD activity, CAT activity, and soluble protein content, and were highly significantly (*p* < 0.01) or significantly (0.01 ≤ *p* < 0.05) negatively correlated with the root biomass, shoot biomass, chlorophyll *a* content, chlorophyll *b* content, carotenoid content, and SOD activity. The root Se content was highly significantly (*p* < 0.01) positively correlated with the shoot Se content.

The root biomass of diploid was highly significantly (*p* < 0.01) or significantly (0.01 ≤ *p* < 0.05) positively correlated with the POD activity, CAT activity, soluble protein content, and TF (Table 4). The shoot biomass was highly significantly (*p* < 0.01) or significantly (0.01 ≤ *p* < 0.05) positively correlated with the CAT activity, soluble protein content, and TF. The root Se content was highly significantly (*p* < 0.01) or significantly (0.01 ≤ *p* < 0.05) positively correlated with the root biomass, carotenoid content, POD activity, and soluble protein content. The shoot Se content was highly significantly (*p* < 0.01) or significantly (0.01 ≤ *p* < 0.05) positively correlated with the root biomass, shoot biomass, chlorophyll *b* content, POD activity, CAT activity, soluble protein content, and TF, and was significantly (0.01 ≤ *p* < 0.05) negatively correlated with the content of chlorophyll *a*/*b*. The root Se content was highly significantly (*p* < 0.01) positively correlated with the shoot Se content.

The root biomass of tetraploid was significantly (0.01 ≤ *p* < 0.05) positively correlated with the chlorophyll *a* content, chlorophyll *b* content, carotenoid content, CAT activity, and soluble protein content (Table 5). The shoot biomass was highly significantly (*p* < 0.01) or significantly (0.01 ≤ *p* < 0.05) positively correlated with the chlorophyll *b* content, POD activity, soluble protein content, and TF. The root Se content and shoot Se content were highly significantly (*p* < 0.01) or significantly (0.01 ≤ *p* < 0.05) positively correlated with the root biomass, shoot biomass, chlorophyll *a* content, chlorophyll *b* content, carotenoid content, POD activity, CAT activity, soluble protein content, and TF. The root Se content was highly significantly (*p* < 0.01) positively correlated with the shoot Se content.

The root biomass of hexaploid was highly significantly (*p* < 0.01) or significantly (0.01 ≤ *p* < 0.05) positively correlated with the chlorophyll *a* content, SOD activity, POD activity, soluble protein content, and TF (Table 6). The shoot biomass was highly significantly (*p* < 0.01) positively correlated with the POD activity, soluble protein content, and TF. The root Se content was highly significantly (*p* < 0.01) or significantly (0.01 ≤ *p* < 0.05) positively correlated with the root biomass, shoot biomass, carotenoid content, POD activity, soluble protein content, and TF. The shoot Se content was highly significantly (*p* < 0.01) or significantly (0.01 ≤ *p* < 0.05) positively correlated with the root biomass, shoot biomass, chlorophyll *a* content, carotenoid content, POD activity, soluble protein content, and TF. The root Se content was highly significantly (*p* < 0.01) positively correlated with the shoot Se content.

### 2.6. Grey Relational Analysis

The grey relationships of different indicators with the shoot Se content were evaluated and the results showed that all indicators were correlated with the shoot Se content (Figure 5). The top four indicators of *C. betacea* with the highest grey correlation coefficients were CAT activity, POD activity, soluble protein content, and root Se content (Figure 5A). The top four indicators with the highest grey correlation coefficients in diploid were POD activity, soluble protein content, TF, and root biomass (Figure 5B). The top four indicators for tetraploid were soluble protein content, POD activity, root Se content, and TF (Figure 5C), and the top four indicators for hexaploid were soluble protein content, shoot biomass, TF, and POD activity (Figure 5D).

## 3. Discussion

Intercropping can change the soil organic acid content, soil pH value, soil nutrient availability, and soil enzyme activity, ultimately affecting the nutrient uptake and growth of plants [22]. In addition, intercropping results in competition between plants; this may inhibit the growth of the plants [23]. Different species of intercropped eggplants exhibited an increase in the biomasses of their seedlings under Se-rich soil conditions [12], whereas intercropping of white radish with green radish resulted in an increase in the biomass of white radish and a decrease in the biomass of green radish [13]. In the current study, *C. betacea* intercropped with sect. *Solanum* showed a decrease in the biomass of *C. betacea* and an increase in the biomass of sect. *Solanum*. This result is consistent with the findings from previous studies [13,14], but different from results from other studies [9,10,12]. This finding indicates competition for growth resources between *C. betacea* and sect. *Solanum*. The biomass of sect. *Solanum* was higher than that of *C. betacea* in the present study. Sect. *Solanum* are herbs, and their growth rates are higher than that of the seedlings of *C. betacea*, a woody plant. Therefore, sect. *Solanum* had a faster growth than *C. betacea*, inhibiting the growth of *C. betacea*. The biomass order of sect. *Solanum* was diploid < tetraploid < hexaploid and intercropping of *C. betacea* also showed that the biomass order was diploid < tetraploid < hexaploid. This finding implies that sect. *Solanum* competed and inhibited the growth of *C. betacea*. The biomass of intercropped sect. *Solanum* showed a positive linear relationship with their biomass of monoculture. The biomass of intercropped *C. betacea* exhibited a linear negative relationship with that of the sect. *Solanum* monoculture. This result further indicates the growth competition between *C. betacea* and sect. *Solanum.* Further studies should be conducted to explore the competition mechanism between the two species.

The content of photosynthetic pigments in plants reflects the photosynthetic capacity [24]. Eggplant intercropped with the Cd-hyperaccumulator *S. nigrum* exhibited a decrease in the photosynthetic pigment content in eggplant planted in Cd-contaminated soil. Tomato intercropped with *S. nigrum* under the same conditions showed an increase in the photosynthetic pigment content in tomato [25]. Grape intercropped with *S. nigrum* planted in Cd-contaminated soil showed increased chlorophyll a and total chlorophyll contents in grape [26]. Maize intercropped with peanut and maize intercropped with soybean showed increased chlorophyll content in maize [7,27]. In the current study, intercropping of *C. betacea* with sect. *Solanum* exhibited a decrease in the photosynthetic pigment content in *C. betacea* and an increase in the photosynthetic pigment content in sect. *Solanum*. These results indicate that the competition between *C. betacea* and sect. *Solanum* inhibited synthesis of photosynthetic pigments in *C. betacea* and promoted the synthesis of photosynthetic pigments in sect. *Solanum*, which may be related to the competition for light, nutrients, and water resources.

High concentrations of Se in plants promote the production of reactive oxygen species (ROS) and induce oxidative stress in plants [28]. Plant defense mechanisms to alleviate Se stress include activating different antioxidant enzymes such as SOD, POD, and CAT [28,29]. Soluble protein is an essential osmotic regulator of plant cells and is implicated in plant stress resistance [30]. Eggplant intercropped with *S. nigrum* in Cd-contaminated soil showed increased antioxidant enzyme activity and soluble protein content of eggplant [31]. Lettuce intercropped with the Cd-hyperaccumulator *G. parviflora* exhibited increased antioxidant enzyme activity and soluble protein content of lettuce [32]. In the present study, intercropping of *C. betacea* with diploid and tetraploid had no significant effects on the SOD activity of *C. betacea*, whereas intercropping of *C. betacea* with hexaploid decreased SOD activity of *C. betacea*. Intercropping *C. betacea* with diploid, tetraploid, and hexaploid increased the POD activity, CAT activity, and soluble protein content of *C. betacea*. Intercropping sect. *Solanum* with *C. betacea* increased the antioxidant enzyme activity and soluble protein content of sect. *Solanum*. These results indicate that intercropping of *C. betacea* with sect. *Solanum* can improve the resistance of the two plants to Se stress, consistent with findings from previous studies [12,13,14]. Moreover, *C. betacea* intercropped with sect. *Solanum* may promote the communication of matters between two plants, resulting in the transfer of the active compounds to each other to mutually improve their resistance to Se stress [33], which needs to be further studied.

The uptake of Se by plants is modulated by various factors such as soil pH value, soil redox conditions, and soil selenium state [4]. Se has a strong metalloid property and exists in inorganic forms with different oxidation states. The different soil oxidation states modulate the various valence states of Se [34,35]. The pathway for absorption of selenite by plants is similar to the absorption pathway for phosphate. Absorption of selenite by plants mainly occurs through the roots [35,36]. On the other hand, there is an antagonism between sulphate and both selenate and selenite forms [37]. Intercropping improves the absorption of nutrients such as nitrogen, phosphorus, and potassium in maize and legumes [7,8,38,39]. Intercropping of heavy metal hyperaccumulator with non-heavy metal hyperaccumulator increases heavy metal uptake in hyperaccumulator but decreases heavy metal uptake in non-hyperaccumulator [9,40]. Intercropping of different genotypes of tomato, eggplant, and radish with themselves exhibits a decrease or increase in Se uptake [12,13,14]. In the current study, intercropping of *C. betacea* with sect. *Solanum* increased the Se uptake in the two plants, consistent with findings on the other plants planted in Cd-contaminated soil [10]. These findings are consistent with results on the uptake levels of Se in tomato, eggplant, and radish reported in previous studies [12,13,14]. The Se contents in intercropped sect. *Solanum* exhibited a positive linear relationship with their Se contents in monoculture. The Se content of intercropped *C. betacea* showed a linear negative relationship with the Se contents in sect. *Solanum* in monoculture. These results indicate that intercropping promoted the Se uptakes in *C. betacea* and sect. *Solanum*, further confirming the communication of matters between *C. betacea* and sect. *Solanum*, and several active compounds may be exchanged between *C. betacea* and sect. *Solanum* to improve the Se accumulation capacity of the two plants. Intercropping of *C. betacea* with diploid increased the TF of *C. betacea*, intercropping with tetraploid had no significant effect on the TF of *C. betacea*, whereas intercropping with hexaploid decreased the TF of *C. betacea*. However, intercropping of sect. *Solanum* with *C. betacea* increased the TFs of sect. *Solanum*. These results indicate that *C. betacea* intercropped with sect. *Solanum* had significant Se effects on sect. *Solanum* than *C. betacea*. This finding explains the higher percentage of shoot Se content in sect. *Solanum* than in *C. betacea*. Correlation and grey relational analyses revealed that CAT activity, POD activity, and soluble protein content were the top three indicators significantly associated with *C. betacea* shoot Se content. In contrast, POD activity, soluble protein content, and TF were the top three indicators markedly associated with sect. *Solanum* shoot Se content. These results further imply that intercropping promoted the Se absorption by increasing the resistance of the two plants to Se, and their action mechanisms should be explored further.

## 4. Materials and Methods

### 4.1. Materials

The *C. betacea* seeds were collected from a five-year-old fruiting tree growing at the Chengdu Campus of Sichuan Agricultural University (30°42′ N, 103°51′ E) in September 2021. The seeds were air-dried. The seeds of *S. photeinocarpum* (diploid) and *S. nigrum* (hexaploid) were obtained from the farmlands around the Chengdu Campus in June 2021. The shoot tips of the diploid seedlings at the stage of unfolded cotyledons were treated with colchicine (0.3%) for 48 h, and the tetraploid was induced. Tetraploid seeds were obtained from the diploid treated with colchicine in September 2021. Diploid, tetraploid, and hexaploid seeds were air-dried. The seeds of four plants were planted in seedling trays filled with wet perlite in February 2022. Hoagland solutions were used to irrigate the seedlings every three days after emergence.

The soil used in this experiment was collected from the farmland around the Chengdu Campus. The soil type was fluvo-aquic soil, and the basic physicochemical properties of soil samples are presented in Table 7 [41].

### 4.2. Experimental Design

The experiment was conducted in a greenhouse at the Chengdu Campus. The soil was treated in March 2022 according to Lin et al. (2020) [42]. Soil with 3.0 kg was placed in a plastic pot (21 cm diameter and 20 cm depth), and Na_2_SeO_3_ (analytical grade) was added to the soil to obtain 5 mg kg^−1^ Se concentration [43]. Then, the soil was watered to maintain the soil moisture content at 80% of the field capacity for one month. In April 2022, uniform plant seedlings were transplanted in the pot. In each pot, four seedlings were planted with even distribution in all four directions. The experiment comprised seven treatments: *C. betacea* monoculture, diploid monoculture, tetraploid monoculture, hexaploid monoculture, *C. betacea* intercropped with diploid, *C. betacea* intercropped with tetraploid, and *C. betacea* intercropped with hexaploid. Four seedlings of one plant species were planted in each pot for the monoculture treatments. Two seedlings of *C. betacea* and two seedlings of each diploid, tetraploid, or hexaploid were planted together in each pot for the intercropping treatments. Each treatment was conducted in triplicate (three pots), and a randomized block design was used in the study. The plants were irrigated using tap water.

### 4.3. Determination of Indicators

To determine whether the intercropping could improve the resistances of these plants to Se, the third mature leaf of each plant from the top was collected to determine various parameters, including the contents of photosynthetic pigments (chlorophyll *a*, chlorophyll *b*, and carotenoid), activities of antioxidant enzymes [superoxide dismutase (SOD), peroxidase (POD), and catalase (CAT)], and soluble protein content following the methods reported by Lin et al. (2023) [44] and Hao et al. (2004) [45] two months after plants transplanting. Subsequently, the plants were harvested and treated as described by Li et al. (2022) [46]. The dry weights (biomass) of roots and shoots were measured using an electronic balance. The plant samples were dried, ground, and digested with nitric acid and perchloric acid. Furthermore, the digestion solutions were reduced with hydrochloric acid, and the Se concentration was determined using a hydride generation-atomic fluorescence spectrometry (AFS-9700, Beijing Haiguang Instrument Co., Ltd., Beijing, China) [46]. The translocation factor (TF, Se content in shoots/Se content in roots) was calculated as described previously [47].

### 4.4. Statistical Analysis

The data were analyzed using SPSS 27.0 software (IBM, Chicago, IL, USA). Data were normalized and subjected to a homogeneity test. One-way analysis of variance was then conducted for comparison of the multiple groups, followed by Duncan’s multiple range test for pairwise comparison (*p* < 0.05). Relationships of the biomass and Se content between the monoculture and intercropping treatments were evaluated using regression analysis. Pearson’s correlation analysis was conducted to determine the correlations among the different indicators of each plant. Grey relational analysis was performed to explore the relationships of the different indicators with the shoot Se content of each plant as described by Wang (2019) [48] and Zhang et al. (2023) [49].

## 5. Conclusions

Intercropping of *C. betacea* with sect. *Solanum* inhibited the growth of *C. betacea* and promoted the growth of sect. *Solanum* as indicated by decrease or increase in their biomasses and photosynthetic pigment contents. In addition, intercropping *C. betacea* with sect. *Solanum* improved their resistances to Se by increasing the levels of antioxidant enzyme activities and soluble protein contents in the two plants. Moreover, intercropping of the two plants increased the shoot Se contents in *C. betacea* and sect. *Solanum*. The biomass and Se content of intercropped sect. *Solanum* exhibited linear relationships with that of monoculture, and the biomass and Se content of intercropped *C. betacea* showed linear relationships with that of sect. *Solanum* monoculture. CAT activity, POD activity, and soluble protein content were the top three indicators highly associated with the shoot Se content in *C. betacea*. The POD activity, soluble protein content, and TF were the top three indicators significantly associated with the shoot Se contents in sect. *Solanum*. Further studies should be conducted under field conditions to verify these findings, and the effects of Se uptake in *C. betacea* fruits should be evaluated.

## Figures and Tables

**Figure 1 plants-12-00716-f001:**
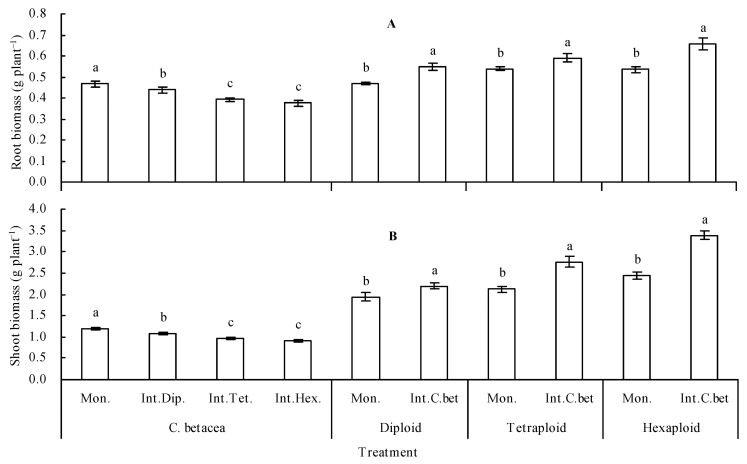
Biomass of plants. (**A**) Root biomass; (**B**) shoot biomass. Values are means ± SD of three replicates. Different lowercase letters indicate significant differences among the treatments (Duncan’s multiple range test, *p* < 0.05). Mon. = monoculture; Int.Dip. = intercropped with diploid; Int.Tet. = intercropped with tetraploid; Int.Hex. = intercropped with hexaploid; Int.C.bet = intercropped with *C. betacea*.

**Figure 2 plants-12-00716-f002:**
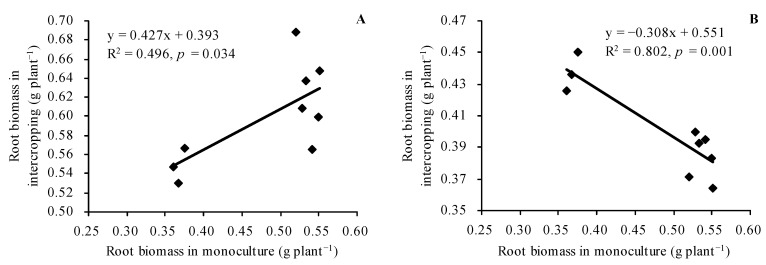
Linear regression relationships of plants’ biomass. (**A**) Root biomasses of diploid, tetraploid, and hexaploid in intercropping plotted against their root biomasses in monoculture; (**B**) root biomasses of *C. betacea* in intercropping plotted against root biomasses of diploid, tetraploid, and hexaploid in monoculture; (**C**) shoot biomasses of diploid, tetraploid, and hexaploid in intercropping plotted against their shoot biomasses in monoculture; (**D**) shoot biomasses of *C. betacea* in intercropping plotted against shoot biomasses of diploid, tetraploid, and hexaploid in monoculture.

**Figure 3 plants-12-00716-f003:**
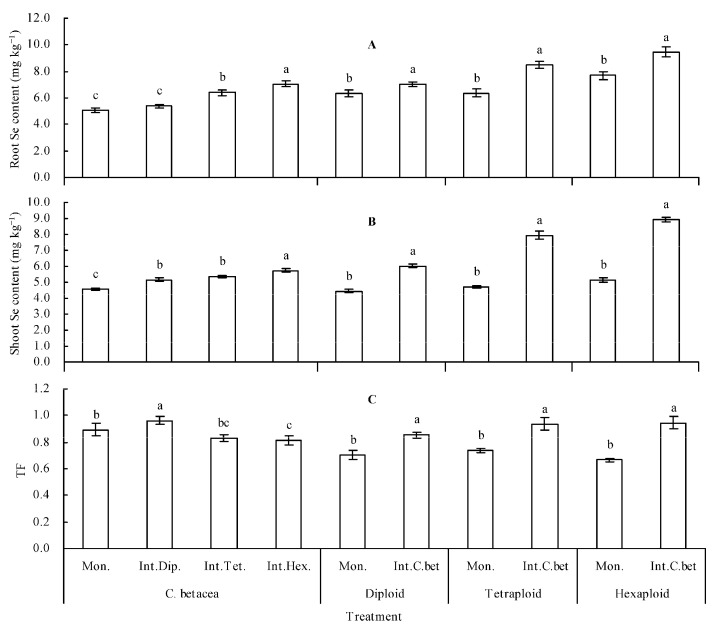
Se content and TF of plants. (**A**) Root Se content; (**B**) shoot Se content; (**C**) TF (translocation factor). Values are means ± SD of three replicates. Different lowercase letters indicate significant differences among the treatments (Duncan’s multiple range test, *p* < 0.05). Mon. = monoculture; Int.Dip. = intercropped with diploid; Int.Tet. = intercropped with tetraploid; Int.Hex. = intercropped with hexaploid; Int.C.bet = intercropped with *C. betacea*.

**Figure 4 plants-12-00716-f004:**
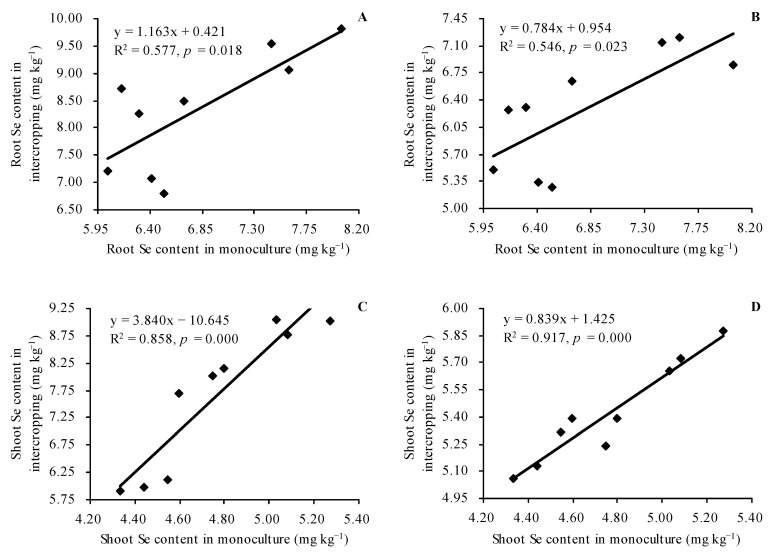
Linear regression relationships of Se contents in plants. (**A**) Root Se contents in diploid, tetraploid, and hexaploid in intercropping plotted against their root Se contents in monoculture; (**B**) root Se contents in *C. betacea* in intercropping plotted against root Se contents in diploid, tetraploid, and hexaploid in monoculture; (**C**) shoot Se contents in diploid, tetraploid, and hexaploid in intercropping plotted against their shoot Se contents in monoculture; (**D**) shoot Se contents in *C. betacea* in intercropping plotted against shoot Se contents in diploid, tetraploid, and hexaploid in monoculture.

**Figure 5 plants-12-00716-f005:**
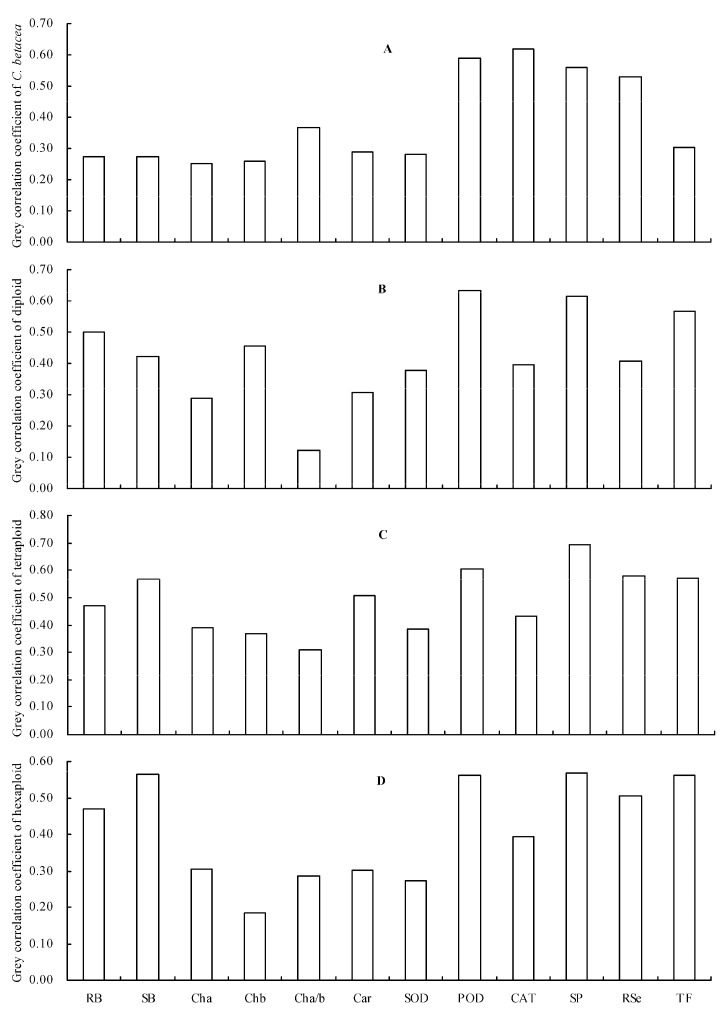
Grey correlation coefficients of the different indicators with the shoot Se content. (**A**) *C. betacea*; (**B**) diploid; (**C**) tetraploid; (**D**) hexaploid. RB = root biomass; SB = shoot biomass; Cha = chlorophyll *a* content; Chb = chlorophyll *b* content; Cha/b = chlorophyll *a*/*b*; Car = carotenoid content; SOD = SOD activity; POD = POD activity; CAT = CAT activity; SP = soluble protein content; RSe = root Se content; TF = translocation factor.

**Table 1 plants-12-00716-t001:** Photosynthetic pigment contents in plants.

Treatment	Chlorophyll *a* Content (mg g^−1^)	Chlorophyll *b* Content (mg g^−1^)	Chlorophyll *a*/*b*	Carotenoid Content (mg g^−1^)
*C. betacea*				
Mono.	1.014 ± 0.047 a	0.445 ± 0.015 a	2.281 ± 0.147 a	0.174 ± 0.002 a
Inter. diploid	0.896 ± 0.031 b	0.399 ± 0.012 b	2.243 ± 0.018 a	0.155 ± 0.004 b
Inter. tetraploid	0.790 ± 0.022 c	0.359 ± 0.014 c	2.203 ± 0.059 a	0.140 ± 0.008 c
Inter. hexaploid	0.789 ± 0.018 c	0.348 ± 0.011 c	2.265 ± 0.023 a	0.135 ± 0.006 c
Diploid				
Mono.	1.668 ± 0.052 a	0.680 ± 0.030 b	2.454 ± 0.059 a	0.285 ± 0.012 a
Inter. *C. betacea*	1.768 ± 0.086 a	0.773 ± 0.035 a	2.288 ± 0.051 b	0.302 ± 0.013 a
Tetraploid				
Mono.	1.873 ± 0.055 b	0.791 ± 0.045 b	2.372 ± 0.078 a	0.309 ± 0.005 b
Inter. *C. betacea*	2.056 ± 0.067 a	0.872 ± 0.018 a	2.358 ± 0.097 a	0.336 ± 0.011 a
Hexaploid				
Mono.	1.976 ± 0.094 b	0.857 ± 0.039 a	2.308 ± 0.144 a	0.322 ± 0.016 b
Inter. *C. betacea*	2.154 ± 0.061 a	0.876 ± 0.023 a	2.459 ± 0.026 a	0.355 ± 0.010 a

Values are means ± SD of three replicates. Different lowercase letters indicate significant differences among the treatments (Duncan’s multiple range test, *p* < 0.05). Mono. = monoculture; Inter. diploid = intercropped with diploid; Inter. tetraploid = intercropped with tetraploid; Inter. hexaploid = intercropped with hexaploid; Inter. *C. betacea* = intercropped with *C. betacea*.

**Table 2 plants-12-00716-t002:** Antioxidant enzyme activities and soluble protein contents of plants.

Treatment	SOD Activity (U g^−1^)	POD Activity (U g^−1^ min^−1^)	CAT Activity (mg g^−1^ min^−1^)	Soluble Protein Content (mg g^−1^)
*C. betacea*				
Mono.	361.1 ± 16.43 a	3889 ± 48.82 c	5.876 ± 0.202 c	12.21 ± 0.56 c
Inter. diploid	353.1 ± 17.71 ab	4397 ± 91.16 b	6.459 ± 0.131 b	13.26 ± 0.47 b
Inter. tetraploid	332.7 ± 10.01 ab	5410 ± 79.98 a	6.585 ± 0.185 b	13.91 ± 0.52 a b
Inter. hexaploid	327.6 ± 13.71 b	5426 ± 58.01 a	7.132 ± 0.191 a	14.56 ± 0.49 a
Diploid				
Mono.	235.1 ± 8.67 b	4666 ± 87.25 b	6.813 ± 0.089 b	10.70 ± 0.20 b
Inter. *C. betacea*	256.8 ± 9.49 a	5771 ± 157.9 a	7.064 ± 0.078 a	12.99 ± 0.58 a
Tetraploid				
Mono.	258.4 ± 9.22 a	5307 ± 102.1 b	7.112 ± 0.095 b	12.18 ± 0.24 b
Inter. *C. betacea*	268.1 ± 5.44 a	7334 ± 316.9 a	7.288 ± 0.049 a	15.94 ± 0.14 a
Hexaploid				
Mono.	272.3 ± 7.78 a	6811 ± 225.3 b	7.484 ± 0.035 a	13.52 ± 0.49 b
Inter. *C. betacea*	283.5 ± 6.54 a	8694 ± 232.2 a	7.542 ± 0.054 a	19.58 ± 0.42 a

Values are means ± SD of three replicates. Different lowercase letters indicate significant differences among the treatments (Duncan’s multiple range test, *p* < 0.05). Mono. = monoculture; Inter. diploid = intercropped with diploid; Inter. tetraploid = intercropped with tetraploid; Inter. hexaploid = intercropped with hexaploid; Inter. *C. betacea* = intercropped with *C. betacea*.

**Table 3 plants-12-00716-t003:** Correlations among the different indicators of *C. betacea*.

Indicator	Root Biomass	Shoot Biomass	Chlorophyll *a* Content	Chlorophyll *b* Content	Chlorophyll *a*/*b*	Carotenoid Content	SOD Activity	POD Activity	CAT Activity	Soluble Protein Content	Root Se Content	Shoot Se Content	TF
Root biomass													
Shoot biomass	0.922 **												
Chlorophyll *a* content	0.880 **	0.953 **											
Chlorophyll *b* content	0.923 **	0.926 **	0.949 **										
Chlorophyll *a*/*b*	0.094	0.315	0.397	0.090									
Carotenoid content	0.893 **	0.942 **	0.918 **	0.952 **	0.129								
SOD activity	0.709 **	0.776 **	0.649 *	0.747 **	−0.107	0.787 **							
POD activity	−0.956 **	−0.957 **	−0.928 **	−0.935 **	−0.217	−0.923 **	−0.738 **						
CAT activity	−0.874 **	−0.838 **	−0.824 **	−0.912 **	0.061	−0.914 **	−0.685 *	0.833 **					
Soluble protein content	−0.779 **	−0.894 **	−0.804 **	−0.830 **	−0.110	−0.858 **	−0.759 **	0.783 **	0.749 **				
Root Se content	−0.912 **	−0.930 **	−0.866 **	−0.865 **	−0.208	−0.841 **	−0.694 *	0.933 **	0.823 **	0.794 **			
Shoot Se content	−0.922 **	−0.930 **	−0.898 **	−0.931 **	−0.127	−0.916 **	−0.733 **	0.881 **	0.936 **	0.855 **	0.870 **		
TF	0.128	0.231	0.254	0.145	0.403	0.246	0.042	−0.236	−0.141	−0.082	−0.190	−0.212	

*n* = 12. **: correlation is significant at the 0.01 level (two-tailed test). *: correlation is significant at the 0.05 level (two-tailed test). TF = translocation factor.

**Table 4 plants-12-00716-t004:** Correlations among the different indicators of diploid.

Indicator	Root Biomass	Shoot Biomass	Chlorophyll *a* Content	Chlorophyll *b* Content	Chlorophyll *a*/*b*	Carotenoid Content	SOD Activity	POD Activity	CAT Activity	Soluble Protein Content	Root Se Content	Shoot Se Content	TF
Root biomass													
Shoot biomass	0.813 *												
Chlorophyll *a* content	0.691	0.531											
Chlorophyll *b* content	0.810	0.739	0.892 *										
Chlorophyll *a*/*b*	−0.733	−0.779	−0.537	−0.859 *									
Carotenoid content	0.705	0.322	0.662	0.706	−0.592								
SOD activity	0.691	0.672	0.332	0.655	−0.827 *	0.315							
POD activity	0.976 **	0.784	0.692	0.869 *	−0.841 *	0.765	0.772						
CAT activity	0.909 *	0.883 *	0.422	0.582	−0.609	0.401	0.651	0.836 *					
Soluble protein content	0.939 **	0.813 *	0.830 *	0.942 **	−0.816 *	0.662	0.759	0.954 **	0.783				
Root Se content	0.888 *	0.552	0.601	0.772	−0.767	0.881 *	0.699	0.945 **	0.675	0.840 *			
Shoot Se content	0.961 **	0.864 *	0.727	0.913 *	−0.883 *	0.692	0.788	0.984 **	0.842 *	0.975 **	0.881 *		
TF	0.896 *	0.955 **	0.711	0.893 *	−0.860 *	0.501	0.759	0.899 *	0.849 *	0.945 **	0.713	0.960 **	

*n* = 6. **: correlation is significant at the 0.01 level (two-tailed test). *: correlation is significant at the 0.05 level (two-tailed test). TF = translocation factor.

**Table 5 plants-12-00716-t005:** Correlations among the different indicators of tetraploid.

Indicator	Root Biomass	Shoot Biomass	Chlorophyll *a* Content	Chlorophyll *b* Content	Chlorophyll *a*/*b*	Carotenoid Content	SOD Activity	POD Activity	CAT Activity	Soluble Protein Content	Root Se Content	Shoot Se Content	TF
Root biomass													
Shoot biomass	0.876 *												
Chlorophyll *a* content	0.823 *	0.784											
Chlorophyll *b* content	0.902 *	0.867 *	0.854 *										
Chlorophyll *a*/*b*	−0.333	−0.326	0.038	−0.487									
Carotenoid content	0.874 *	0.788	0.963 **	0.788	0.114								
SOD activity	0.563	0.613	0.308	0.298	−0.011	0.492							
POD activity	0.787	0.968 **	0.803	0.769	−0.105	0.807	0.596						
CAT activity	0.865 *	0.739	0.946 **	0.847 *	−0.037	0.929 **	0.209	0.734					
Soluble protein content	0.835 *	0.958 **	0.855 *	0.782	−0.043	0.881 *	0.646	0.987 **	0.780				
Root Se content	0.937 **	0.960 **	0.880 *	0.873 *	−0.178	0.907 *	0.566	0.945 **	0.880 *	0.963 **			
Shoot Se content	0.863 *	0.948 **	0.913 *	0.832 *	−0.047	0.922 **	0.571	0.972 **	0.849 *	0.992 **	0.975 **		
TF	0.736	0.888 *	0.895 *	0.747	0.082	0.882 *	0.545	0.950 **	0.766	0.969 **	0.897 *	0.972 **	

*n* = 6. **: correlation is significant at the 0.01 level (two-tailed test). *: correlation is significant at the 0.05 level (two-tailed test). TF = translocation factor.

**Table 6 plants-12-00716-t006:** Correlations among the different indicators of hexaploid.

Indicator	Root Biomass	Shoot Biomass	Chlorophyll *a* Content	Chlorophyll *b* Content	Chlorophyll *a*/*b*	Carotenoid Content	SOD Activity	POD Activity	CAT Activity	Soluble Protein Content	Root Se Content	Shoot Se Content	TF
Root biomass													
Shoot biomass	0.932 **												
Chlorophyll *a* content	0.870 *	0.805											
Chlorophyll *b* content	0.534	0.359	0.463										
Chlorophyll *a*/*b*	0.597	0.646	0.793	−0.172									
Carotenoid content	0.770	0.764	0.484	−0.008	0.540								
SOD activity	0.815 *	0.683	0.972 **	0.598	0.669	0.345							
POD activity	0.891 *	0.968 **	0.726	0.155	0.699	0.883 *	0.572						
CAT activity	0.456	0.586	0.312	−0.377	0.607	0.756	0.128	0.723					
Soluble protein content	0.957 **	0.991 **	0.803	0.374	0.634	0.810	0.691	0.969 **	0.626				
Root Se content	0.939 **	0.893 *	0.784	0.234	0.707	0.921 **	0.681	0.939 **	0.656	0.929 **			
Shoot Se content	0.957 **	0.983 **	0.824 *	0.317	0.695	0.839 *	0.706	0.980 **	0.597	0.985 **	0.956 **		
TF	0.921 **	0.994 **	0.811	0.337	0.668	0.754	0.686	0.964 **	0.543	0.975 **	0.886 *	0.983 **	

*n* = 6. **: correlation is significant at the 0.01 level (two-tailed test). *: correlation is significant at the 0.05 level (two-tailed test). TF = translocation factor.

**Table 7 plants-12-00716-t007:** The basic chemical properties of soil.

Soil Type	pH Value	Organic Matter Content (g kg^−1^)	Total N Content (g kg^−1^)	Total P Content (g kg^−1^)	Total K Content (g kg^−1^)	Alkaline Hydrolyzed N Content (mg kg^−1^)	Available P Content (mg kg^−1^)	Available K Content (mg kg^−1^)	Total Se Content (mg kg^−1^)
Fluvo-aquic	7.71	15.29	1.85	11.88	15.38	87.99	55.78	41.96	0.12

## Data Availability

Data will be made available on genuine request.

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
