# Peer review of "Intercropping of *Cyphomandra betacea* with Different Ploidies of *Solanum* Sect. *Solanum* (Solanaceae) Wild Vegetables Increase Their Selenium Uptakes"

_plants, 2023, doi:10.3390/plants12040716_

Round 1
Reviewer 1 Report
The paper was interesting up to a point. I have some oncerns
line 16 the indicators did not increase - their values did
CAT POD and SOD should be defined at first mention - most readers would not know what you mean. The paper does not explain why these were measured - what is the question you are trying to answer by measuring them?
lies 43-45 - this is written as a universal principle which it is not. Further into the paragraph you show why it is not.
Line 85 + Only one source of seed and only one environment - how do you know that it isn't a biotype? Can you be confident that findings can be extrapolated to other seed sources?
Line 89+ - why are you looking at ploidy - is it because you can or is there an hypothesis?
line 105 - "uniform seedlings of plant seedlings" needs correction.
Lines 107 - 111 - treatments could be describe in a table for clarity
Line 113 - why not a randomised block design rather than completely randomised - justification needed.
line 148 and 149 "root biomasses .....were...."
Nowhere has the interaction of selenium uptake with sulfur uptake been mentioned. A paper by Pratley and McFarlane (1972) Australian J of Experimental Agriculture and Animal Husbandry 14, 5333-538 describes this interaction and is likely to be a factor here.
Tables 4,5 and 6 headings need to be sorted so the first word doesn't go to the second line (e.g Chlorophyll, Carotenoid)
Introduction and discussion seem very similar and cover the same information. The introduction needs to be clearer on what the questions being investigated are.
The discussion has introduced allelopathy but not the word into the discussion. Competition is also discussed for the first time in any detail. These are likely very important influences. Rhizosphere dialogue was introduced but not expanded but rather postulated. Finally the term resistance to Se was introduced but not described.
It is very hard to be definitive on what is happening between roots and in the rhizosphere without knowing what the changes are in the chemical composition of the exudates - exudates were mentioned but no data were forthcoming.
So what did the paper achieve? Did selenium levels reach satisfactory? Were uptake levels excess? if they increase were there deleterious impacts on other nutrients e.g. sulfur? What is the value of changing ploidy to get higher selenium levels? What is the basis for studying SOD, CAT, POD in terms of selenium levels?
While the title of the paper was about selenium levels, selenium formed only a minor part of the paper and the reader does not know why.
It is only one experiment with one interrow species - is this experiment repeatable. I am unsure about that.
Author Response
Comments and Suggestions for Authors
The paper was interesting up to a point. I have some oncerns
Thank you for your reviewing.
line 16 the indicators did not increase - their values did
We have added “values”.
CAT POD and SOD should be defined at first mention - most readers would not know what you mean. The paper does not explain why these were measured - what is the question you are trying to answer by measuring them?
We have added the full name of CAT, POD, and SOD in Abstract, and in Materials and Methods.
Determines of CAT, POD, and SOD activities are explaining whether the intercropping could improving the resistances of these plants to Se. We have added the description in Materials and Methods.
lies 43-45 - this is written as a universal principle which it is not. Further into the paragraph you show why it is not.
We have revised as “Intercropping can improve the rate of utilization of light, temperature, water, fertilizer and other resources by crops to some extent. Moreover, intercropping also can modulate crop root exudates, soil pH value, and soil enzyme activity, ultimately improving crop yield and quality to some extent [4,6]”.
Line 85 + Only one source of seed and only one environment - how do you know that it isn't a biotype? Can you be confident that findings can be extrapolated to other seed sources?
The C. betacea has no cultivars, only has local variety with self-pollination. The C. betacea tree used collecting seeds is an individual plant. So, it is a biotype.
Line 89+ - why are you looking at ploidy - is it because you can or is there an hypothesis?
Solanum photeinocarpum and Solanum nigrum are annual to perennial Solanum sect. Solanum wild vegetables with a high Se accumulation capacity. S. photeinocarpum is a diploid plant, and S. nigrum is a hexaploid plant, whereas S. nigrum is evolved from S. photeinocarpum in nature condition. The tetraploid S. photeinocarpum is rarely found in nature. So, we used colchicines inducing S. photeinocarpum (tetraploid). This information is in Introduction.
line 105 - "uniform seedlings of plant seedlings" needs correction.
We have revised as “uniform plant seedlings”.
Lines 107 - 111 - treatments could be describe in a table for clarity
We have tried to use the table, but the table looks ugly.
Line 113 - why not a randomised block design rather than completely randomised - justification needed.
We wrote it wrong. This is a one-factor randomized block design. We have revised.
line 148 and 149 "root biomasses .....were...."
We have revised.
Nowhere has the interaction of selenium uptake with sulfur uptake been mentioned. A paper by Pratley and McFarlane (1972) Australian J of Experimental Agriculture and Animal Husbandry 14, 5333-538 describes this interaction and is likely to be a factor here.
We have added in Discussion section.
Tables 4,5 and 6 headings need to be sorted so the first word doesn't go to the second line (e.g Chlorophyll, Carotenoid)
Tables 4,5 and 6 have been re-arranged.
Introduction and discussion seem very similar and cover the same information. The introduction needs to be clearer on what the questions being investigated are.
We have revised.
The discussion has introduced allelopathy but not the word into the discussion. Competition is also discussed for the first time in any detail. These are likely very important influences. Rhizosphere dialogue was introduced but not expanded but rather postulated. Finally the term resistance to Se was introduced but not described.
We have revised these information in Discussion.
It is very hard to be definitive on what is happening between roots and in the rhizosphere without knowing what the changes are in the chemical composition of the exudates - exudates were mentioned but no data were forthcoming.
We have deleted them.
So what did the paper achieve? Did selenium levels reach satisfactory? Were uptake levels excess? if they increase were there deleterious impacts on other nutrients e.g. sulfur? What is the value of changing ploidy to get higher selenium levels? What is the basis for studying SOD, CAT, POD in terms of selenium levels?
The aim of study was determine whether the combination of sect. Solanum and C. betacea could improve Se accumulation in these plants, and screened the best combination. The paper has achieved this aim, and the Se levels reach satisfactory. For C. betacea, the Se in its fruits is lower than seedlings. In this experiment, the Se content is calculated as dry weight, but the fruits and wild vegetables are used fresh to eat. The Se content in fresh is lower than dry. So, uptake levels are not excess.
Se uptake may deleterious impact on other nutrients e.g. sulfur, which need to be further studied.
Solanum photeinocarpum and Solanum nigrum are nature wild vegetables, which can be planted under C. betacea tree for some economic and food values.
The SOD, CAT, POD activities are all directly related to plant Se accumulation, which directly affects the Se uptake in plants. So these indicators were measured.
While the title of the paper was about selenium levels, selenium formed only a minor part of the paper and the reader does not know why.
The physiological indicators, including the photosynthetic pigments, activities of antioxidant enzymes, and soluble protein content, are all directly related to plant Se accumulation, which directly affects the Se uptake in plants. So these indicators were measured.
It is only one experiment with one interrow species - is this experiment repeatable. I am unsure about that.
This is a preliminary study. Next, we will be conducted under field conditions to verify these findings, and the effects of Se uptake in C. betacea fruits will be evaluated. The further studies have been written in CONCLUSIONS.
Reviewer 2 Report
Dear Authors
Do you measure Cd? How can you conclude about Cd resistance in conclusion? Please rewrite the conclusion based on your finding.
please arrange the correlation tables.
Author Response
Comments and Suggestions for Authors
Dear Authors
Do you measure Cd? How can you conclude about Cd resistance in conclusion? Please rewrite the conclusion based on your finding.
Thank you for your reviewing. It should be Se. We have revised. The conclusion is also rewritten.
please arrange the correlation tables.
The correlation tables have been re-arranged.
Reviewer 3 Report
All comments are inserted in text.

Author Response
Comments and Suggestions for Authors
All comments are inserted in text.
Thank you for your reviewing. We have revised the paper according to your comments in the PDF file. The unit of SOD activity (U g−1) is right.